# Synthesis of Highly Porous Graphene Oxide–PEI Foams for Enhanced Sound Absorption in High-Frequency Regime

**DOI:** 10.3390/polym16212983

**Published:** 2024-10-24

**Authors:** Seung-Chan Jung, Wonjun Jang, Byeongji Beom, Jong-Keon Won, Jihoon Jeong, Yu-Jeong Choi, Man-Ki Moon, Eou-Sik Cho, Keun-A Chang, Jae-Hee Han

**Affiliations:** 1Department of Materials Science and Engineering, Gachon University, Seongnam 13120, Republic of Korea; jsc4486@gachon.ac.kr (S.-C.J.); juni1205@gachon.ac.kr (W.J.); jjkk3239@gachon.ac.kr (B.B.); wjk0508@gachon.ac.kr (J.-K.W.); wlgns9751@gachon.ac.kr (J.J.); dbwjd330@gachon.ac.kr (Y.-J.C.); wangpeterpan@gachon.ac.kr (M.-K.M.); 2Department of Electronic Engineering, Gachon University, Seongnam 13120, Republic of Korea; es.cho@gachon.ac.kr; 3Department of Pharmacology, College of Medicine, Gachon University, Incheon 21999, Republic of Korea; keuna705@gachon.ac.kr

**Keywords:** graphene oxide–polyethyleneimine (GO-PEI) foam, high-frequency noise, sound absorption, porous structure, acoustic properties

## Abstract

High-frequency noise exceeding 1 kHz has emerged as a pressing public health issue in industrial and occupational settings. In response to this challenge, the present study explores the development of a graphene oxide–polyethyleneimine (GO-PEI) foam (GPF) featuring a hierarchically porous structure. The synthesis and optimization of GPF were carried out using a range of analytical techniques, including Raman spectroscopy, scanning electron microscopy (SEM), Braunauer–Emmett–Teller (BET) analysis, X-ray diffraction (XRD), and Fourier-transform infrared spectroscopy (FT-IR). To evaluate its acoustic properties, GPF was subjected to sound absorption tests over the 1000–6400 Hz frequency range, where it was benchmarked against conventional melamine foam. The findings demonstrated that GPF with a GO-to-PEI composition ratio of 1:3 exhibited enhanced sound absorption performance, with improvements ranging from 15.0% to 118%, and achieved a peak absorption coefficient of 0.97. Additionally, we applied the Johnson–Champoux–Allard (JCA) model to further characterize the foam’s acoustic behavior, capturing key parameters such as porosity, flow resistivity, and viscous/thermal losses. The JCA model exhibited a superior fit to the experimental data compared to traditional models, providing a more accurate prediction of the foam’s complex microstructure and sound absorption properties. These findings underscore GPF’s promise as an efficient solution for mitigating high-frequency noise in industrial and environmental applications.

## 1. Introduction

In recent years, the accelerated pace of urbanization and the expansion of transportation infrastructure have presented significant environmental challenges, with noise pollution emerging as a pressing concern in contemporary society [1]. A growing body of research has linked noise exposure to a range of adverse health outcomes, including cardiovascular disorders, cognitive impairments, sleep disturbances, tinnitus, and general discomfort [2,3]. According to studies and reports by the World Health Organization (WHO), prolonged exposure to noise at frequencies around 1 kHz poses substantial risks to auditory health, with high-frequency sound in this range potentially leading to long-term hearing impairment or loss [4]. In a landmark study involving 790 aircraft manufacturing workers exposed to noise levels between 4 and 6 kHz, it was revealed that workers in the highest noise exposure categories faced a 1.46- to 1.48-fold increased risk of developing hypertension, underscoring the significant health risks associated with high-frequency noise [5]. These findings not only highlight the dangers to auditory health but also point to broader systemic health implications, emphasizing the urgency of addressing noise pollution in industrial settings. As industrialization continues to advance, exposure to noise from various sources, particularly from motors used in transportation and domestic environments, remains pervasive. Market projections suggest that the internal combustion engine and electric motor industries will experience significant growth by 2030 and 2027, respectively, further exacerbating this issue [6,7]. In light of these developments, there is a critical need for advanced sound-absorbing materials capable of addressing high-frequency noise, typically ranging from 1000 to 6400 Hz. High-frequency noise, commonly generated by industrial machinery, electronic devices, and transportation systems, presents a unique challenge due to its higher energy levels and its ability to penetrate conventional sound barriers.

This study presents an innovative approach to sound absorption by synthesizing graphene oxide (GO) via the Hummers method [8,9], introducing functional groups that are subsequently cross-linked with branched polyethyleneimine (PEI) in an aqueous solution. Ultrasonic enhancement was employed to facilitate the dispersion of GO sheets within the PEI matrix, reinforcing chemical bonding and promoting uniform cross-linking [10,11,12,13,14]. This bottom-up synthesis approach results in a polymer matrix with improved homogeneity, stability, and sound absorption properties, particularly in the high-frequency range. Raman spectroscopy and photoluminescence analyses were employed to determine the optimal ratio of GO to PEI in the hydrogel [15]. Following this, the hydrogel was rapidly cooled using liquid nitrogen, a process that traps moisture and leads to the formation of both micro- and nano-sized pores during the subsequent freeze-drying [16,17]. The rapid cooling process is essential for moisture entrapment, while the freeze-drying step enhances porosity, contributing to the material’s capacity to absorb a broad range of sound frequencies. The interaction between the carboxyl groups of GO and the amine groups of PEI results in a robust pore structure, capable of multi-frequency absorption [18,19]. These pore structures, crucial for absorbing a diverse range of frequencies, were confirmed via scanning electron microscopy (SEM) imaging, which revealed the presence of both nano- and micro-sized pores. Braunauer–Emmett–Teller (BET) analysis further validated the increased surface area, which plays a pivotal role in sound wave interaction and absorption [20]. Additionally, Fourier-transform infrared spectroscopy (FT-IR) and X-ray diffraction (XRD) revealed strong amine and carboxyl cross-linking, contributing to the material’s mechanical stability and sound absorption efficiency. The developed GO-PEI foam (GPF) composite was rigorously tested across a frequency range of 1000 to 6400 Hz, demonstrating superior sound absorption performance, particularly in the high-frequency regime, when compared to conventional materials. The synergy between the diverse pore size distribution and the enhanced structural stability achieved through ultrasonic processing and freeze-drying underscores the potential of this GPF as a highly effective solution for mitigating noise pollution in environments characterized by high-frequency noise.

## 2. Materials and Methods

### 2.1. Materials

Graphite powder (<20 μm) and branched polyethyleneimine (Mw ~25,000) were obtained from Sigma-Aldrich (Saint Louis, MO, USA). Sulfuric acid (95–98 wt%) was produced by JT. Baker (Radnor, PA, USA). Hydrochloric acid (35–37 wt%) and Phosphoric acid (85 wt%) were purchased from Duksan (Seoul, Republic of Korea). Potassium permaganate (99.3 wt%) and ethanol were produced by SAMCHUN (Seoul, Republic of Korea). Diethyl ether (99.7 wt%) was obtained from CARLO ERBA (Milano, Italy). Polyethyleneimine (average Mw ~25,000, branched) was obtained from Sigma-Aldrich. Melamine foam (pore size, 100–150 μm) was obtained from Dong Sung Co. (Seoul, Republic of Korea).

### 2.2. Graphene Oxide Synthesis

GO was synthesized using the modified Hummers method [8,9]. We used H_2_SO_4_ and H_3_PO_4_ (9:1 *v*/*v* ratio) for intercalating the solution for graphite layers. The addition of KMnO_4_ generated thermal energy, which exfoliated the graphite layers and introduced functional groups such as those found in GO. After the above step, the washing process was conducted to enable a stable neutralization process by removing residual ions. We performed centrifugation for 15 min in the conditions of 4500 rpm, 4 °C. The above process was repeated by changing the washing solution in the order of deionized (DI) water, ethanol, and diethyl ether. The samples that underwent the process were dried in a vacuum state to become flakes.

### 2.3. Graphene Oxide–PEI Porous Foam Synthesis

The first step involves the formation of a GO-PEI hydrogel. First, a mixture of GO dispersion (1 mg/mL) in aqueous solution and a polyethyleneimine (PEI) dispersion was prepared. We used tip sonication (BRANSON Digital Sonifier Model S—450) to ensure the uniform dispersion of GO and PEI [10,11,21,22]. Ultrasonic cavitation generated localized heating and kinetic energy, promoting the bonding between the carboxyl groups of GO and the amine groups of PEI, facilitating C-N bond formation [23]. Sonication was performed at 10% amplitude for 15 min to ensure effective dispersion. We prepared samples with varying mass ratios of GO to PEI (1:1, 1:2, 1:3, and 1:4). These ratio conditions aim to optimize the interaction between GO and PEI, based on the sp^2^-hybridized structure of GO. The samples were rapidly frozen using liquid nitrogen to preserve the porous structure and then freeze-dried using a freeze dryer (FDU-1200, EYELA).

### 2.4. Instrumental Characterization

The difference in bonding structure between GO and GO-PEI foam was distinguished through Raman spectroscopy with excitation wavelength of 532 nm (HORIBA Jobin Yvon NanoLog, HORIBA Scientific, Kyoto, Japan). Afterwards, it was measured using FT-IR (iS50, Thermo Fisher Scientific, Waltham, MA, USA) to confirm the bonding that occurred as GO-PEI was synthesized. XRD (Smartlab, Rigaku, Tokyo, Japan) was additionally used to confirm the synthesis of a composite during this process. Afterwards, in order to confirm the pore size of the optimized foam, imaging using SEM and inspection using BET (ASAP 2020 Micromerities, Norcross, GA, USA) were conducted together. In the case of the sound absorption performance test, the impedance measurement within the tube was conducted by Korea Apparel Testing & Research Institute (KATRI) (Cheongju, Republic of Korea).

### 2.5. Sound Absorption in an Impedance Tube

Sound absorption happens when sound waves encounter a material in their path. As the waves strike the material, they can be reflected, absorbed, or remain within the material for some time. When the waves pass through, their kinetic energy is transformed into heat as they interact with the pores of the material [24,25].
(1)$$\alpha \;(\mathrm{sound}\;\mathrm{absorption}\;\mathrm{coefficient})=\frac{I_{absorbed}}{I_{incident}}$$

In the case of the sound absorption coefficient, it is expressed as the ratio of *I_absorbed_* (absorbed sound energy value) to *I_incident_* (incident sound energy value) in the above formula.
(2)$$\alpha \;(\mathrm{sound}\;\mathrm{absorption}\;\mathrm{coefficient})=1-{\left|R\right|}^{2}$$
(3)$$R=\frac{P_{r}}{P_{i}}$$

For the sound absorption coefficient in an impedance tube, it is expressed through *R* (reflected energy ratio). At this time, *R* is expressed as the ratio of *P_r_* (reflected sound pressure) to *P_i_* (incident sound pressure).

## 3. Results and Discussion

### 3.1. Characterization and Sound Absorption Properties of Highly Porus GO-PEI

Figure 1 presents the Raman spectroscopy results for graphite, GO, and GPF samples with varying GO-to-PEI ratios. Each sample was analyzed by measuring the D and G bands using a 532 nm laser. Across all samples, the D band is located at 1342 cm⁻^1^ and the G band at 1568 cm⁻^1^ [10]. Graphene, characterized by its sp^2^-hybridized structure, can be oxidized to introduce functional groups that enable interactions in sp^3^-hybridized sites. During this process, the D peak intensifies, indicating an increase in the intensity ratio of the D band to G band (*I_D_*/*I_G_*). The right-hand side of Figure 1 compares these ratios across the samples. The progressive rise in the *I_D_*/*I_G_* ratio, reaching a value of 1.1 as GO is combined with PEI, signifies the formation of functional groups, likely resulting from successful GO synthesis. However, when the GO-PEI ratio exceeds 1:3, the *I_D_*/*I_G_* ratio decreases to 1.04, indicating that further PEI addition beyond this point does not facilitate more interactions in the sp^3^-hybridized positions of GO. This suggests that excess PEI remains unreacted, pointing to a saturation threshold in the functionalization process. This observed increase in the *I_D_*/*I_G_* ratio with PEI functionalization has also been reported in previous studies. For example, a Raman analysis of GO functionalized with various polymers, including PEI, has consistently demonstrated an upward shift in the *I_D_*/*I_G_* ratio, confirming an increase in defect formation on the carbon skeleton as sp^3^-hybridized sites are functionalized. As described in Refs. [26,27], similar observations were made, where the *I_D_*/*I_G_* ratio showed notable increases after PEI conjugation, further supporting the activation of bonding interactions in the GO-PEI system. This finding aligns with our results and reinforces the conclusion that the *I_D_*/*I_G_* ratio can serve as a reliable indicator of functionalization extent in GO-based composites.

Table 1 shows snapshots and SEM images of each hydrogel and GPF formed through freeze-drying as PEI was incrementally added to GO. As the PEI content increases from 1:1 to 1:3 in GPF, the presence of gold on the surface is noticeably reduced. However, at higher PEI ratios, such as 1:4 and 1:30, cracks begin to form on the surface again. The SEM images display well-formed composites in the samples with 1:1 to 1:3 ratios, but at the 1:4 ratio, the structure shows signs of bulking, and in the 1:30 samples, the composite characteristics observed at lower ratios are lost, with pores ranging in size from hundreds of micrometers to several millimeters. This suggests that beyond 1:3, additional PEI leads to excess bulking rather than effective bonding with GO.

Additionally, a comparative analysis of the *I_D_*/*I_G_* ratio (as shown in Figure 1) and the SEM images of freeze-dried GPF supports this observation. Beyond the 1:3 ratio, the D band intensity relatively decreases, surface cracks appear, and the pore size changes, as seen in the SEM images. These findings indicate that the number of functional groups available in the sp^3^-hybridized region of GO is limited, and once a certain PEI mass is exceeded, it negatively impacts the creation of pores, leading to bulking and structural instability.

Figure 2 presents a comparative analysis of the BET isotherm linear plots for GPF 1:3 and GPF 1:4, highlighting their adsorption and desorption behaviors. These plots provide key insights into the internal structure and pore distribution within the samples. The BJH adsorption average pore diameter for the GPF 1:3 sample is 7.25 nm, while GPF 1:4 exhibits a larger average diameter of 15.34 nm. For desorption, the average pore diameter is 5.13 nm for GPF 1:3 and 14.33 nm for GPF 1:4 [28]. These values indicate that as the GO-PEI ratio increases beyond 1:3, the number of smaller pores decreases, resulting in an overall increase in pore size. In the GPF 1:3 plot, a small amount of gas is adsorbed at low relative pressure, with adsorption steadily increasing as the relative pressure rises. The hysteresis loop formed by the divergence between the adsorption and desorption curves suggests that gas adsorption and desorption within the material are not entirely reversible, confirming the porous nature of the material [29,30].

In contrast, the GPF 1:4 plot reveals larger pore sizes, with gas adsorption starting at low pressure. The desorption curve is relatively lower at low pressures compared to the adsorption curve but recovers as the pressure increases. This phenomenon is attributed to the higher concentration of PEI in the GPF 1:4 sample, which coats and blocks some of the surface pores, making them behave like closed pores [31,32]. Based on changes observed in the *I_D_*/*I_G_* ratio in Raman spectroscopy, SEM, and BET analyses (refer to Figure 1), it is evident that when PEI exceeds a certain amount, the excess PEI blocks the pores, increases the average pore size, and compromises the physical properties of the material [31]. Consequently, GPF 1:3 is determined to have the optimal ratio for maintaining a balanced porous structure and superior material performance. Additionally, for the GPF 1:1 and 1:2 samples, BET measurements could not be performed. This was due to significant challenges in achieving stable foam formation at these ratios, which resulted in large and small cracks forming throughout the materials, as shown in Table 1 for the corresponding samples. These structural defects rendered the samples unsuitable for reliable BET analysis, as the cracks compromised the overall integrity of the foam.

Since the manufacturing process of GPF relies on the bonding between the functional groups of GO and PEI, the formation of and shifting in peaks observed through FT-IR are crucial for understanding the resulting structure [23,26,27]. In Figure 3, GO (indicated by the blue dashed line), it is formed by oxidizing graphene, which results in prominent peaks corresponding to C-O, C=O, and O-H groups due to oxidation. In contrast, GPF 1:3 exhibits distinct peaks that reflect the bonding between GO and PEI. Specifically, the red dashed lines in the FT-IR graph in Table 2 represent characteristic peaks formed by this bonding. Notably, the O-H peak around 3400 cm^−1^ is significantly reduced, indicating that the functional groups of GO are being utilized in bonding with PEI. Furthermore, the red dashed lines corresponding to C-N stretching (1120 cm^−1^) and N-H bending (1570 cm^−1^) highlight the formation of these bonds.

Figure 4 presents an XRD graph that highlights the structural differences between GO and GPF 1:3. Unlike FT-IR, which provides insight into chemical bonds and their states, XRD is used to analyze the crystallinity and layered structure of materials [33,34]. The XRD pattern for GO, represented by the black solid line, shows a peak at 9.68°. This peak corresponds to the layered structure of GO and indicates the distance between layers. This interlayer distance is influenced by oxygen-containing functional groups, such as carboxyl, hydroxyl, and epoxy groups. In the XRD pattern of GPF (represented by the red solid line), the peak shifts to 22.67°, and the peak of FWHM (Full-width at Half Maximum) increases from 0.95 to 7.72 compared to that of GO. Based on Bragg’s law, this peak shift suggests that the insertion of PEI between GO layers reduces the interlayer spacing due to the formation of C-N and N-H bonds [35]. Additionally, the increase in the half-width of the peak can be explained using the Scherrer equation.
(4)nλ=2dsinθ

In Bragg’s law, n refers to the reflection order, *λ* is the wavelength of the X-ray, *d* is the interlayer spacing, and *θ* is the diffraction angle. Therefore, as the 2*θ* value increases, the d-spacing decreases, which indicates a reduction in the interlayer distance. In Figure 4, the theta value for graphene oxide (GO) is 9.68°, while for graphene oxide–polyethyleneimine (GO-PEI). it is 22.67°. This increase in the θ value corresponds to a reduction in the interlayer distance of GO-PEI by approximately 59.9%.
(5)$$D=\frac{K\lambda }{\;\beta cos\theta }$$

Formula (5) represents the Scherrer equation, where *D* denotes the crystal size, *K* is the Scherrer constant, λ is the X-ray wavelength, and *β* and *θ* refer to the half-width of the peak and diffraction angle, respectively [36]. Thus, the increase in the half-width of the peak in GPF, caused by the interaction between GO and PEI, indicates that the material is being reorganized into a nanostructure by reducing the functional groups. Based on the FWHM and *θ* values observed in the XRD analysis, a comparison using the Scherrer equation reveals that the crystal size decreased by approximately 87.5%.

Figure 5 illustrates the sound absorption mechanism of GPF 1:3, developed through careful material analysis and optimization to respond effectively to high-frequency noise. When sound waves enter the GPF material, the kinetic energy is transformed into heat as it interacts with the nano-sized pores created by graphene oxide (GO) and the micro-sized pores formed by the combination of GO and polyethyleneimine (PEI). This interaction results in an efficient sound absorption process where the heat generated is radiated away. The Barrett–Joyner–Halenda (BJH) adsorption and desorption curves in Figure 5 further confirm the presence of both nano- and micro-sized pores in the GPF 1:3 structure [37]. The pore distribution analysis indicates that nanopores larger than 2 nm are uniformly distributed, ensuring an effective surface area for sound absorption. Additionally, the SEM images provide visual evidence of a well-structured porous network with micro-sized pores clearly visible, further substantiating the material’s capacity for sound absorption through the intricate porous architecture. These combined analyses demonstrate that GPF 1:3 is well suited for dissipating high-frequency sound energy, leveraging both nanoscale and microscale porosity to achieve high sound absorption efficiency.

Figure 6 presents a comprehensive analysis of sound absorption performance for GPF 1:3 and melamine foam, based on impedance tube measurements carried out by the Korea Apparel Testing and Research Institute (KATRI) [38]. Additional details on images of the sample molds used for GPF and melamine foams for sound absorption measurement are represented in Appendix A. The frequency range under investigation spans from 1000 Hz to 6400 Hz, highlighting how GPF 1:3 significantly outperforms melamine foam in terms of sound absorption across different frequency bands. In the left graph, the sound absorption coefficient is plotted against frequency, showing that GPF 1:3 consistently performs better than melamine foam, particularly in the low-to-mid-frequency range (1000 Hz to 4000 Hz). GPF 1:3 exhibits superior absorption, with coefficients approaching 0.97 at 5000 Hz and maintaining an impressive average value of 0.92 across the entire range. By comparison, melamine foam shows a slower increase in absorption capacity, especially in the lower frequencies, where GPF demonstrates a distinct advantage. The right-side bar chart quantifies the percentage improvement in GPF 1:3 over melamine foam, revealing a striking 118% increase in the 1000–2000 Hz range. This dramatic enhancement in low-frequency sound absorption is critical for many practical applications, including noise reduction in environments where low-frequency sound dominates, such as industrial or building settings. Additionally, the graph highlights a 44.8% improvement in the 2000–4000 Hz range and a 15% increase in the higher 4000–6400 Hz range, further supporting GPF’s effectiveness as a high-performance soundproofing material. The results of these tests indicate that GPF 1:3 not only provides a more efficient absorption of high-frequency noise compared to melamine foam, but it also excels in the critical low- and mid-frequency ranges. This enhanced performance makes GPF a strong candidate for applications where comprehensive acoustic management is required, solidifying its potential as a superior alternative to conventional soundproofing materials like melamine foam.

### 3.2. Theoretical Modeling for Acoustic Properties of Highly Porous GO-PEI Foams Using JCA, Delany–Bazley, and Miki Models

In this study, we utilized the Johnson–Champoux–Allard (JCA) model to characterize the acoustic properties of our GO-PEI foam (GPF 1:3). The JCA model, which accounts for additional parameters such as porosity, flow resistivity, tortuosity, and viscous/thermal characteristic lengths, was employed to capture the complex behavior of sound absorption in this material [39,40,41,42,43]. As shown in Figure 7, our experimental results demonstrate a close fit between the JCA model predictions (blue dashed line) and the actual sound absorption coefficients, particularly in the mid-to-high-frequency range of 1000–6400 Hz. The Delany–Bazley and Miki models, traditionally used for simpler porous materials, were also applied to the GPF for comparison [44,45]. However, as demonstrated in Figure 7, these models (green and red dashed lines) significantly underpredict the sound absorption behavior, especially at higher frequencies, where the foam’s intricate microstructure plays a dominant role. This is further supported by the poor statistical performance of these models, with the coefficient of determination *R*^2^ values approaching −65. The JCA model, on the other hand, provides a significantly better fit, with an *R*^2^ value of approximately 0.5, indicating a much more reliable prediction of the acoustic properties of GPF. The optimized JCA model not only accounts for material characteristics such as porosity and tortuosity but also incorporates the effects of viscous and thermal losses within the foam. Additional data on the theoretical modeling is represented in Appendix A. These microstructural factors significantly influence the acoustic impedance, as described by the following formulas:

Dynamic Effective Density (*ρ_eff_* (*ω*)):(6)$$\rho_{eff}\left(\omega \right)=\rho_{0}\phi \alpha_{\infty }\left(1+\frac{\sigma \phi }{i\omega \rho_{0}\alpha_{\infty }}\sqrt{1+\frac{4i{\alpha_{\infty }}^{2}\rho_{0}\omega }{\sigma^{2}\Lambda^{2}\phi^{2}}}\right)$$
where *ρ*_0_ is the air density, ϕ is the porosity, *σ* is the flow resistivity, and *α_∞_* is the high-frequency tortuosity. This equation accounts for the viscous effects that dominate at high frequencies, particularly in materials with complex pore structures like GPF.

Dynamic Bulk Modulus (*K_eff_* (*ω*)):(7)$$\frac{1}{K_{eff}(\omega )}=\frac{\gamma -(\gamma -1){\left(1+\frac{8N_{pr}ik\phi }{\sigma N^{2}\rho_{0}c_{p}\omega }\sqrt{1+\frac{4i{\alpha_{\infty }}^{2}\rho_{0}N_{pr}\omega }{\sigma^{2}\Lambda^{2}\phi^{2}}}\right)}^{-1}}{P_{0}}$$
where *c_p_* is the specific heat capacity of air, and *Λ* and *Λ′* are the viscous and thermal characteristic lengths. This formulation captures the thermal losses that occur due to the foam’s intricate pore network, particularly at lower frequencies.

Acoustic Impedance:(8)$$Z\left(\omega \right)=\frac{\rho_{eff}(\omega )c_{eff}(\omega )}{\phi }$$

Formula (8) represents the acoustic impedance *Z*(*ω*) from the Johnson–Champoux–Allard (JCA) model. The impedance *Z*(*ω*) is crucial for determining the foam’s sound absorption behavior across the frequency spectrum. The correlation between acoustic impedance and the sound absorption coefficient is given by the following:(9)$$\alpha \left(\omega \right)=1-{\left|\frac{Z\left(\omega \right)-Z_{0}}{Z\left(\omega \right)+Z_{0}}\right|}^{2}$$
where *Z*_0_ is the characteristic impedance of air.

As illustrated in Figure 6, the sound absorption performance of GPF 1:3 was analyzed over a broad frequency range, from 1000 Hz to 6400 Hz. The foam exhibited enhanced absorption in the 2000–4000 Hz range, showing a 44.8% improvement over conventional soundproofing materials like melamine foam. Additionally, the GPF outperformed in the 4000–6400 Hz range, achieving a 15% increase in sound absorption, further demonstrating its effectiveness for high-performance soundproofing applications. This result aligns well with the JCA model’s predictions, confirming that the material’s microstructure, particularly its tortuosity and viscous losses, play critical roles in enhancing sound absorption at higher frequencies. The JCA model captures the nuanced acoustic behavior of the foam, as shown by the close match between the experimental data (black squares) and the model’s fit (blue dashed line) in Figure 7.

## 4. Conclusions

This study highlights the increasing concern over high-frequency noise and the critical role porous materials can play in addressing this challenge. In response, we developed a porous foam based on a nano-graphene composite formed through the interaction of graphene oxide (GO) and polyethyleneimine (PEI). This foam exhibited strong structural integrity and exceptional sound absorption capabilities, particularly in the frequency range of 1000 to 6400 Hz, where the GPF 1:3 composition achieved a maximum sound absorption coefficient of 0.97. These characteristics make the foam well suited for noise control in applications such as brushless DC motors, electric vehicles, and industrial machinery where high-frequency noise management is essential [46]. In addition to experimental characterization using the Raman, SEM, BET, XRD, and FT-IR techniques, we employed the Johnson–Champoux–Allard (JCA) model to further understand the foam’s acoustic properties. The JCA model, which considers key parameters such as porosity, flow resistivity, and viscous and thermal losses, provided a strong fit to the experimental data, particularly at mid-to-high frequencies. Compared to the Delany–Bazley and Miki models, the JCA model demonstrated superior predictive accuracy, especially for materials like GPF with complex internal pore structures. The Delany–Bazley and Miki models significantly underperformed in these higher frequency ranges, as evidenced by poor *R*^2^ values near −65. In contrast, the JCA model achieved a much more reliable *R*^2^ value of approximately 0.5, reinforcing its ability to accurately capture the foam’s acoustic behavior. The optimized JCA model not only provided a more accurate understanding of the foam’s sound absorption properties but also offered insights into the microstructural factors—such as tortuosity and thermal losses—that significantly influence performance. This makes the JCA model an invaluable tool for the design and optimization of advanced soundproofing materials, particularly those engineered for high-frequency noise environments.

Moving forward, a further refinement of GO-PEI foam synthesis could enhance its performance even further, and the potential to expand its application to lower-frequency noise control should be explored. Overall, this study provides a foundation for future developments in nanomaterial-based soundproofing technologies, paving the way for more effective noise mitigation solutions in a variety of industrial and commercial settings.

## 5. Patents

Porous foam structure for high-frequency sound absorption using graphene oxide, sound absorbing material including the same and manufacturing method thereof [KR Patent Application No. 10-2022-0155236].

## Figures and Tables

**Figure 1 polymers-16-02983-f001:**
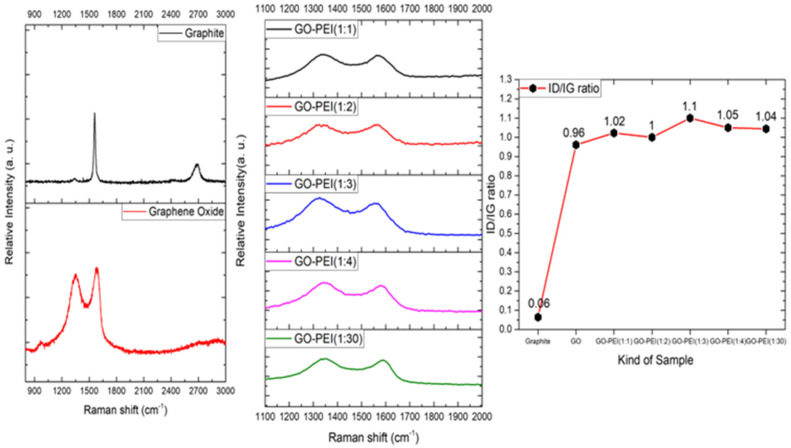
Raman spectroscopy analysis of graphite and GO as well as GPFs with different ratios of GO to PEI within them.

**Figure 2 polymers-16-02983-f002:**
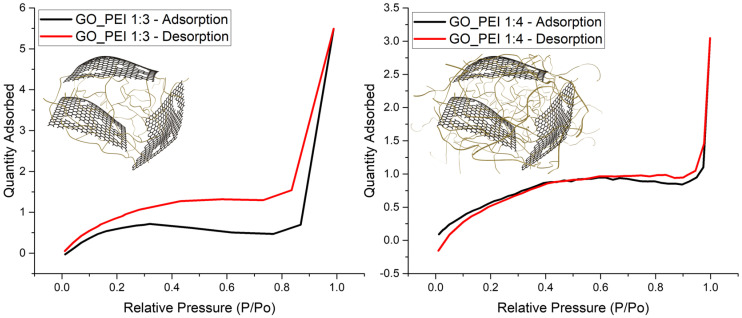
BET isotherm linear plot and schematic of porous structures in GPF 1:3 and GPF 1:4.

**Figure 3 polymers-16-02983-f003:**
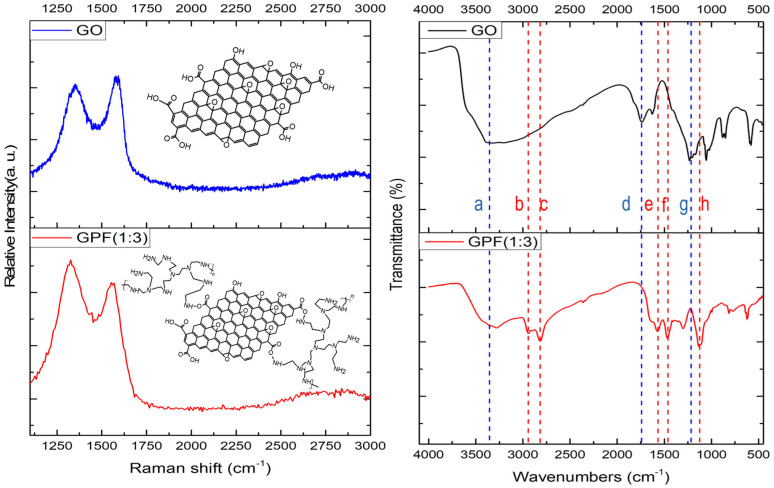
Comparative Raman (**left** panels) and FT-IR spectra (**right** panels) of GO and GPF 1:3 samples.

**Figure 4 polymers-16-02983-f004:**
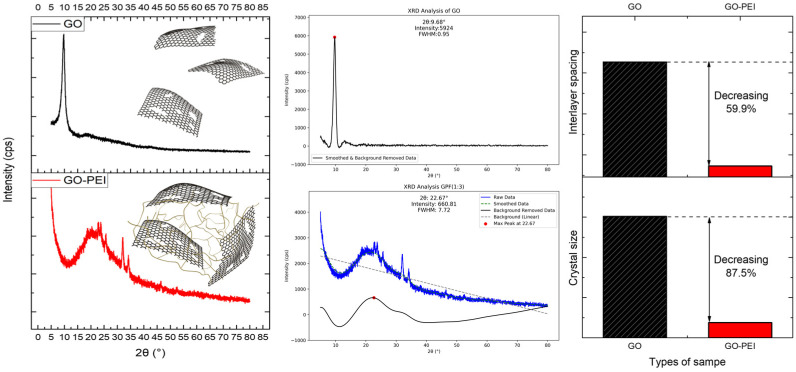
XRD-based comparative analysis and structural schematics of GO and GPF 1:3 samples.

**Figure 5 polymers-16-02983-f005:**
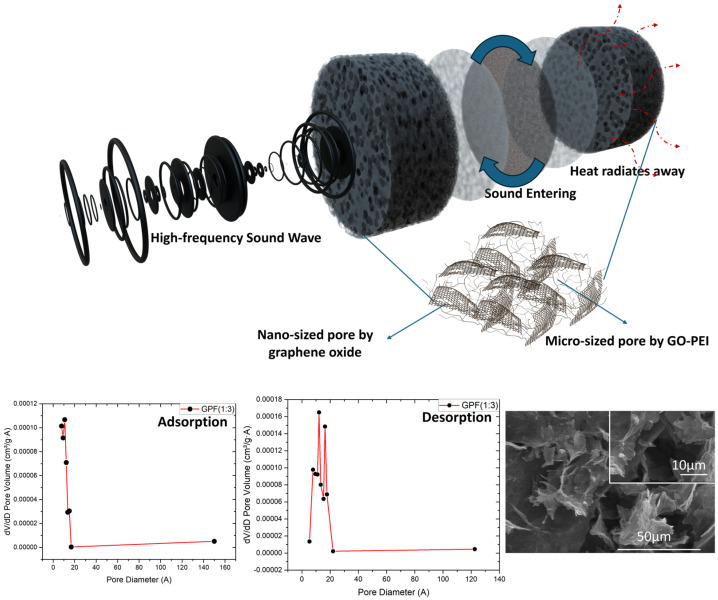
Schematic of sound absorption mechanism in GPF 1:3 and pore size distribution analyzed via BJH and SEM.

**Figure 6 polymers-16-02983-f006:**
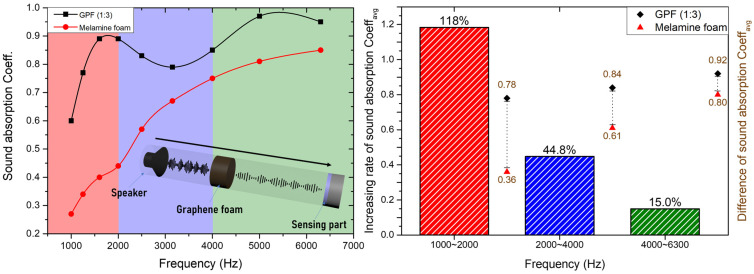
Schematic representation and analysis of sound absorption rates for GPF 1:3 and melamine foam.

**Figure 7 polymers-16-02983-f007:**
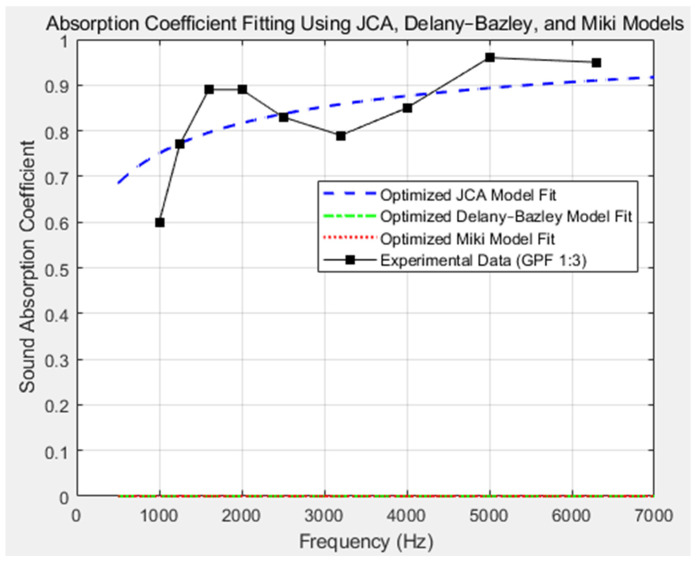
A comparison of sound absorption coefficients for GPF 1:3 and melamine foam, fitted using the Johnson–Champoux–Allard (JCA), Delany–Bazley, and Miki models. The experimental data for GPF 1:3 are shown with black square markers, highlighting the superior fitting accuracy of the JCA model (blue dashed line) compared to the Delany–Bazley (green dashed line) and Miki (red dotted line) models across the frequency range of 1000–6400 Hz.

**Table 1 polymers-16-02983-t001:** Snapshots and SEM images of GO-PEI hydrogel and foam structures formed by different ratios of GO to PEI within each GPF composite.

GO-to-PEI Ratio in GPF (*w*/*w*)	Snapshot of Hydrogel	Snapshot of Foam	SEM Image of Foam
GPF 1:1	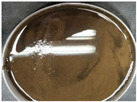	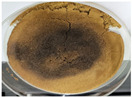	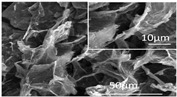
GPF 1:2	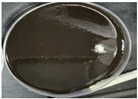	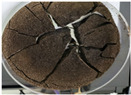	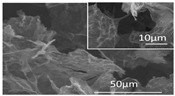
GPF 1:3	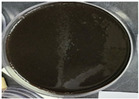	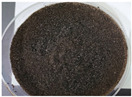	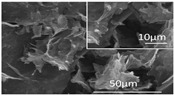
GPF 1:4	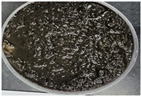	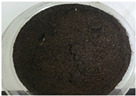	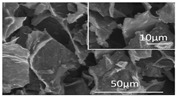
GPF 1:30	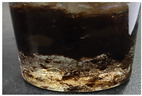	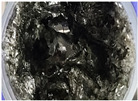	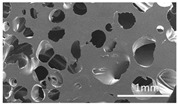

**Table 2 polymers-16-02983-t002:** FT-IR analysis of bonding states in both GO and GPF 1:3 samples.

Wavenumbers (cm^−1^)	Peak of GO (Blue Dashed Line)	Peak of GPF 1:3 (Red Dashed Line)	Types of Bonding
(a) 3400	O-H stretching	N-H and O-H stretching	O-H and N-H stretching (hydroxyl of GO and amine of PEI)
(b) 2950	-	C-H stretching (asymmetric)	C-H stretching (asymmetric) (alkyl chains of PEI)
(c) 2800	-	C-H stretching (symmetric)	C-H stretching (symmetric) (alkyl chains of PEI)
(d) 1720	C=O stretching	-	C=O stretching (carbonyl group in GO)
(e) 1570	-	N-H bending C-N stretching	N-H bending, C-N stretching (GO interacting with PEI)
(f) 1460	-	C-H bending	C-H bending (vibration from alkyl groups in PEI)
(g) 1120	-	C-N stretching	C-N stretching (C-N bonds from GPF)
(h) 1060	C-O stretching (relatively strong)	C-O stretching (relatively weak)	C-O stretching (oxidized groups from GO)

## Data Availability

The original contributions presented in the study are included in the article, further inquiries can be directed to the corresponding author.

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
