# Peer review of "Synthesis of Highly Porous Graphene Oxide–PEI Foams for Enhanced Sound Absorption in High-Frequency Regime"

_polymers, 2024, doi:10.3390/polym16212983_

Round 1
Reviewer 1 Report
Comments and Suggestions for Authors
1. In Figure 1 the small and erratic change of ID/IG ratio between 0.96 and 1.1 could not suggest the formation of functional group between GO and PEI since D and G bands in Raman spectra represent the disordered carbon part and graphitic carbon part. Only FT-IR results could validate the chemical linkage of C-N bond.
2. It is better to supply with BET plots of GPF (1:1) and GPF (1:2) in Figure 2 to show the trend.
3. The formats of Table 1 and Table 2 are irregular and also the FT-IR curves should be shown separately from the Table 2.
Author Response
Comments No. 1: In Figure 1 the small and erratic change of ID/IG ratio between 0.96 and 1.1 could not suggest the formation of functional group between GO and PEI since D and G bands in Raman spectra represent the disordered carbon part and graphitic carbon part. Only FT-IR results could validate the chemical linkage of C-N bond.
Our response No. 1: By referencing Fig. 4 of Ref. [1] and Fig. 1 of Ref. [2], we found that monitoring changes in the ID/IG ratio in Raman spectra for both GO and GO-polymer composites is an effective method for detecting bonding and phase transitions in these materials. Furthermore, as shown by previous studies [2–5], the combined application of Raman spectroscopy alongside other analytical techniques, such as FT-IR, XPS, and XRD, provides a comprehensive approach to confirm the formation of C-N bonding. This integrated analysis is particularly advantageous, as changes in specific Raman peaks can reliably indicate material interactions, while a specific analysis like FT-IR offers deeper insights into the chemical nature of these interactions. The synergistic use of Raman spectroscopy with complementary techniques such as FT-IR thus provides a robust and powerful framework for studying material interfaces, as highlighted in earlier research.
The gradual increase in the ID/IG ratio observed in the Raman spectra, as the PEI content increases in GO-PEI composites, likely suggests C-N bond formation. However, as Reviewer #1 correctly pointed out, drawing immediate conclusions about C-N bond activation solely based on changes in the ID/IG ratio requires further supporting analysis such as FT-IR.
To address this, we revised the original manuscript to emphasize that the interaction between GO and PEI is likely facilitated by the addition of PEI, as indicated by the changes in the ID/IG ratio presented in Figure 1 of the revised manuscript. Furthermore, we have added a new Figure 3, which now includes both Raman spectroscopy and FT-IR analysis results for the GO and GPF (1:3) samples. Consequently, the numbering of the original Figure 3 and all subsequent figures has been adjusted accordingly. This integrated approach, validated by previous studies, provides a more logical and effective way to present our findings. We hope our response in this issue addresses the reviewer's comments appropriately, and we sincerely thank you for the valuable comments.
[Ref [1], M Jannatin et al 2019 IOP Conf. Ser.: Earth Environ. Sci. 217 012007. https://doi.org/10.1088/1755-1315/217/1/012007]
“The intensity ratio of D band and G band (ID/IG) is very susceptible to structural changes of carbon matrix which can be affected by several factors such as doping, defect, substrate, and so on. D band relates to imperfections of carbon structures which reflected the structural defects and associated with part of the sp3 carbon atom obtained due to graphite amorphization during the oxidation process while the G band is derived from vibrations in the sp2 carbon atomic plane. There was an ID/IG ratio increase on the GO before and after being GO-nano Fe3O4 composite of 0.79 to 1.45. Previous research has also produced similar changes [17]. It was caused by the n-type effect on the GO that made the sp2 carbon change to sp3 so that the intensity ratio of D band and G band increased. In other words, there has been a change of crystal structure from pure GO to GO-nano Fe3O4 composite.”
[Ref [2], Journal of Nanoscience and Nanotechnology, 14(10), 7395–7401. https://doi.org/10.1166/jnn.2014.9571]
“The ID/IG ratio changed after GO was reacted with 6-arm-PEG-NH2. The ID/IG ratio of GO was 0.916, but that of GO-PEG-NH2 was 1.028. This transition of the ID/IG ratio indicates a possible decrease in the average size of the sp2 domains because of chemical changes (Fig. 1(c)).”
[Ref [3], ACS Appl. Nano Mater. 2021, 4, 8637−8640. https://doi.org/10.1021/acsanm.1c01645]
“Next, EDA/GO was subjected to Raman analysis, which was used to calculate the ID/IG ratio. The ID/IG ratio provides information regarding the structural properties of the GO where the intensity of the D band (ID) is indicative of the extent of disorder in the basal plane, while the G band (IG) arises from the ordered sp2-hybridized domains. We observed a considerable increase in the ID/IG ratio from 0.92 for GO to 1.03 for EDA/GO (Figure 1b), which supports the substitution of oxygen-containing functional groups with the less-electronegative nitrogen functional group (−NH−R) from EDA. A similar increase in the ID/IG ratio has been previously reported for EDA-stitched GO sheets.”
[Ref [4], Small, 10(1), 117–126. https://doi.org/10.1002/smll.201202636]
“Raman analysis also provided conclusive evidence of GO reduction. A change of value in the intensity ratio between the D and G peaks from 0.97 to 1.13 was observed after BPEI conjugation and reduction (Figure 1b). [25] The conjugation ratio of PEG to BPEI in PEG–BPEI–rGO is found to be 0.8 (molar ratio) as determined by 1H NMR analysis (Figure S1).”
[Ref [5], Abbas, Y. (2018). Surface acoustic waves (SAW) mediated crumpled polyethyleneimine-labelled graphene oxide (CGO-PEI) synthesis as nanotherapeutic delivery platform (Master's thesis). https://researchrepository.rmit.edu.au/esploro/outputs/9921864161101341]
“For carbonaceous materials, Raman spectroscopy is considered a powerful tool for their characterization. Thus, to ensure maintenance of GO features after nebulization via SAWs and functionalization with PEI, Raman spectroscopy was used. The Raman spectra of CGO and CGO-PEI obtained (Fig 12) show two peaks at 1339 and 1609 cm-1 for CGO, 1335 and 1599 cm-1 for CGO-PEI; respectively where each peak corresponds to sp3 carbon atoms of the defect structure (D-band) and sp2-hybridized carbon atoms from the aromatic structure of GO (G-band); respectively [136, 137]. By calculating the ID/IG ratio for each spectrum the ratio for CGO-PEI which is 1.37 is higher than that of CGO which has a value of 1.19 indicating increase in the defect on the carbon skeleton from functionalization with PEI.”
Comments No. 2: It is better to supply with BET plots of GPF (1:1) and GPF (1:2) in Figure 2 to show the trend.
Our response No. 2: Thank you for your valuable comment and for providing additional guidance. As highlighted in Table 1, the GPF (1:3) and GPF (1:4) samples demonstrated a well-formed foam structure, which enabled us to successfully carry out BET measurements. In contrast, the GPF (1:1) and GPF (1:2) samples, which contain lower PEI content, exhibited significant challenges in achieving stable foam formation. Specifically, the optimization of the composition in these samples was not sufficient to prevent the formation of structural defects, leading to the presence of both large and small cracks throughout the material. These cracks have rendered the samples unsuitable for BET analysis due to compromised structural integrity, making it impossible to acquire reliable data.
We sincerely appreciate your suggestion and have addressed this limitation in our discussion. Furthermore, we have added this explanation regarding the BET measurement challenges for the GPF (1:1) and GPF (1:2) samples in the revised manuscript. (page 6, line 210~215) We hope that our clarification resolves the concerns associated with the BET measurements for these samples.
Comments No. 3: The formats of Table 1 and Table 2 are irregular and also the FT-IR curves should be shown separately from the Table 2.
Our response No. 3: We have separated the FT-IR graph previously included in Table 2 and presented it independently as the new Figure 3. This new figure also incorporates the Raman spectroscopy results for GO and GPF (1:3), as shown in the revised manuscript. Additionally, we have resized Table 1 and Table 2 to comply with the formatting guidelines provided by MDPI. (page 5, Table 1) (page 6, Figure 3) (page 7, Table2) Thank you once again for your grateful attention and feedback.
References for comments of Reviewer #1
- M Jannatin et al 2019 IOP Conf. Ser.: Earth Environ. Sci. 217 012007. https://doi.org/10.1088/1755-1315/217/1/012007
- Byun, E., & Lee, H. (2014). Enhanced loading efficiency and sustained release of doxorubicin from hyaluronic acid/graphene oxide composite hydrogels by a mussel-inspired catecholamine. Journal of nanoscience and nanotechnology, 14(10), 7395–7401. https://doi.org/10.1166/jnn.2014.9571
- ACS Appl. Nano Mater. 2021, 4, 8637−8640. https://doi.org/10.1021/acsanm.1c01645
- Kim, H., & Kim, W. J. (2014). Photothermally controlled gene delivery by reduced graphene oxide-polyethylenimine nanocomposite. Small, 10(1), 117–126. https://doi.org/10.1002/smll.201202636
5. Abbas, Y. (2018). Surface acoustic waves (SAW) mediated crumpled polyethyleneimine-labelled graphene oxide (CGO-PEI) synthesis as nanotherapeutic delivery platform (Master's thesis). German University in Cairo. https://researchrepository.rmit.edu.au/esploro/outputs/9921864161101341

Reviewer 2 Report
Comments and Suggestions for Authors
Please review the attached file for reviewed details.

Author Response
[Reviewer #2]
The authors have proposed a porous graphene oxide-polyethyleneimine foam to treat sound noise with frequency above 1kHz. The proposed graphene foam shows improved sound absorption performance in the targeted frequency range. However, high frequency (>1kHz) sound noise treatment has never been being a big challenge for acoustic community, thanks to its relatively small wave-length and weak penetration feature. Still, there are some interesting directions remaining for dedicated investigation, especially those related to improving noise treatment efficiency (lightweight, thin, and less materials), effectiveness, durability, anti-humidity performance, etc. In general, the proposed acoustic absorption material could be interesting to acoustic community, even though the acoustic theory part needs to be further explored. Several critical questions/suggestions listed below need to be addressed before it can be further considered for the next step.
Comments No. 1: The acoustic model needs to be explored. It could be more rigorous if the authors can provide acoustic impedance of the proposed material.
Our response No. 1: We would like to express our sincere appreciation for Reviewer #2’s valuable feedback, which has allowed us to steer our research in a more refined direction. In general, acoustic models can be defined using a theoretical approach based on materials. In Fig. 2 of Ref [1] seen just below, the models such as Delany-Bazley, MiKi, and Johnson-Champoux-Allard (JCA) are commonly used.
[Ref [1], Polym. Test. 2021, 104, 107388. https://doi.org/10.1016/j.polymertesting.2021.107388]
To address the Reviewer #2’s comments regarding the acoustic modeling of our graphene oxide-polyethyleneimine (GO-PEI) foam, we have employed the Johnson-Champoux-Allard (JCA) model to capture the material's acoustic performance more accurately. In addition to the JCA model, we compared our results with the Delany-Bazley and Miki models to evaluate their predictive capacity for our foam [2-8].
Sound absorption coefficient versus acoustic impedance:
The relation between the sound absorption coefficient α(ω) and the acoustic impedance Z(ω) is given by Equation (1):
|
|
Equation 1. Correlation between acoustic impedance and sound absorption coefficient
Where Z0 is the characteristic impedance of air.
Optimized JCA model fit:
The JCA model, which accounts for additional material properties such as porosity Φ, tortuosity α∞​, viscous characteristic length Λ, and thermal characteristic length Λ′, provided a better fit to the experimental data with optimized fitting parameters in Table 1, as shown in Figure 1. This model is more appropriate for materials like our GO-PEI foam, where these microstructural parameters significantly affect acoustic behavior. The dynamic effective density and bulk modulus of the material, central to the JCA model, are represented by Equations (2) to (4), respectively:
- Dynamic Effective Density (ρeff(ω)):
Equation 2. Dynamic Effective Density
ρ0: Air density (about 1.21 kg/m3)
Ï•: Porosity
α∞: High-frequency tortuosity
σ: Flow resistivity (Pa·s/m²)
ω: Angular frequency
i: Imaginary unit
Λ: Viscous characteristic length (m)
- Dynamic Bulk Modulus (Keff(ω)):
Equation 3. Dynamic Bulk Modulus
γ: Adiabatic index of air (approximately 1.4)
P0: Atmospheric pressure (approximately 101325 Pa)
Npr: Prandtl number (approximately 0.71)
k: Thermal conductivity of air (approximately 0.026 W/m·K)
cp: Specific heat capacity of air (approximately 1005 J/kg·K)
Λ′: Thermal characteristic length (m)
- Acoustic Impedance (Z(ω)):
Equation 4. Acoustic impedance Z(ω) from JCA (Johnson-Champoux-Allard) model.
The optimized JCA model fit is depicted by the blue dashed line in Figure 1 seen below, showing a clear match with the experimental data (black square markers) across the frequency range of interest, particularly for mid-to-high frequencies where the foam's complex microstructure plays a dominant role. These results suggest that the JCA model captures the material's acoustic behavior with greater accuracy than the other two models.
Figure 1. Comparison of sound absorption coefficient for GPF (1:3) foam based on non-linear fitting using the Johnson-Champoux-Allard (JCA), Delany-Bazley, and Miki models. The optimized JCA model (blue dashed line) shows a closer fit to the experimental data (black squares) compared to the Delany-Bazley (green dashed line) and Miki (red dotted line) models, which fail to accurately predict the absorption behavior of the foam across the frequency range.
Table 1. Optimized Johnson-Champoux-Allard (JCA) model parameters for the GO-PEI foam, including flow resistivity (σ), tortuosity (α∞​), porosity (Φ), viscous characteristic length (Λ), and thermal characteristic length (Λ′), derived from fitting the experimental sound absorption data. (Table 1 has been included in Supplementary Materials of the revised manuscript)
|
Flow Resistivity (σ) |
55.7E+01 |
|
Tortuosity (α∞) |
2.71E+00 |
|
Porosity (Φ) |
6.09E-02 |
|
Viscous Char. Length (Λ) |
3.44E-05 |
|
Thermal Char. Length (Λ′) |
2.14E-05 |
Comparison with Delany-Bazley and Miki models:
For comparison, we also applied the Delany-Bazley and Miki models (also seen above in Figure 1), which are traditionally used for simpler porous materials. These models, which mainly rely on flow resistivity σ, provided significantly poorer fits to the experimental data, as shown by the green dashed line (Delany-Bazley) and the red dotted line (Miki) in the graph. As indicated in the statistical comparison (Figure 2), the coefficient of determination R2 values for these models were approximately -65, indicating that they are inadequate for capturing the nuanced acoustic behavior of our GO-PEI foam.
Figure 2. Statistical comparison of model fits for the JCA, Delany-Bazley, and Miki models, summarizing the Root Mean Square Error (RMSE), Mean Absolute Error (MAE), and Coefficient of Determination R² values for each model. (Figure S2 has been included in Supplementary Materials of the revised manuscript)
While the JCA model provided a more accurate fit than the Delany-Bazley and Miki models, it is important to acknowledge that even the JCA model does not fully account for the intricate microstructure of the foam, as reflected by the moderate R2 value of approximately 0.5. The complexity of the foam’s pore structure, including irregularities in pore shapes and non-uniform pore size distributions, likely introduces additional mechanisms of sound absorption that are not captured by the standard JCA model formulation. This suggests that further refinement of the model or the inclusion of additional physical effects may be necessary to fully describe the acoustic performance of materials with such complex internal structures.
Key insights from the optimization process for acoustic modeling of GPF foams using JCA, Delany-Bazley, and Miki models:
(1) Parameter sensitivity: Flow resistivity σ and porosity Φ have the most significant influence on the absorption behavior across the frequency spectrum. Tortuosity α∞​ and thermal characteristic length Λ′ play critical roles in the mid-to-high frequency range, where the internal paths of the foam and thermal losses contribute to the overall absorption performance.
(2) JCA model superiority: The JCA model, through its incorporation of viscous and thermal characteristic lengths, provides a superior framework for modeling the acoustic properties of our foam, especially when compared to simpler models like Delany-Bazley and Miki.
In conclusion, the JCA model optimization results, combined with our experimental validation, demonstrate that our foam’s acoustic properties are reasonably well-characterized, particularly for the high-frequencies regime ranging from 1000-6400 Hz. However, the moderate R2 value underscores the need for potential refinements to the model to fully capture the foam’s complex microstructure. The comparison with the Delany-Bazley and Miki models further emphasizes the limitations of those models in dealing with materials of this complexity. We believe these findings provide valuable insights into the acoustic performance of materials with intricate pore structures, such as our GO-PEI foam, and can guide future studies focused on noise treatment materials with similarly complex internal architectures. Furthermore, we have included these insights and the JCA model analysis in the revised manuscript to provide a comprehensive explanation of the foam’s acoustic performance and the modeling techniques used. (page 10, line 308 to page 12, line 370)
Comments No. 2: Since the high frequency is not a critical issue in noise reduction. For instance, an ordinary sponge material could have a very good acoustic performance (see the green curve in the figure below, from ‘Optimal Sound-Absorbing Structures’ Mater.)
Our response No. 2: We sincerely appreciate your insightful comments on our manuscript and recognize the critical importance of addressing low-frequency noise, which has been extensively documented in the literature. However, we would like to draw attention to the potential health hazards associated with high-frequency noise, which are also highlighted in several referenced studies. For instance, experimental research has demonstrated significant health risks linked to prolonged exposure to high-frequency noise. Specifically, Ref [9] (along with the other references listed at the end of this Response letter) reports pathological changes in the brains and hearts of rats continuously exposed to 4000 Hz noise. Additionally, studies focusing on human subjects have illustrated the direct health impacts of high-frequency noise. Ref [10] documents a correlation between high-frequency noise exposure and increased cardiovascular risks, including a notable rise in hypertension incidents at frequencies around 4 kHz. Moreover, due to advancements in transportation technologies and the resulting shifts in societal conditions, research into the detection of sounds up to 6400 Hz has continued up until recently [11]. Taken together, these findings suggest that high-frequency noise warrants as much research attention as low-frequency noise, due to its potentially detrimental health effects.
In this regard, our study, which centers on the high-frequency range of 1000-6400 Hz, seeks to contribute to this important, yet underexplored, area of research. We are grateful for the opportunity to compare our results with those of the sponge material discussed in the reference suggested by Reviewer #2, which demonstrated a great performance in the 0-3000 Hz range. As recommended by Reviewer #2, we acknowledge the importance of highlighting our work's strength in addressing a frequency range that is not often covered in existing studies. However, we find it challenging to make a direct comparison with the referenced sponge material, as specific structural characteristics, such as pore size and distribution, are not provided in the paper, which limits the feasibility of a comprehensive analysis.
Nonetheless, our thorough literature review has identified several materials that exhibit promising acoustic absorption properties within the high-frequency regimes. Moreover, as stated in our comments, existing limitations in sound-absorbing materials are often addressed through innovative design methodologies [12-14]. Our study aims to develop sound-absorbing materials with tunable pore structures and sizes, achieved through dehydration-condensation reactions between two-dimensional GO and branched PEI containing multiple amine groups. By fine-tuning the ratios of these components, we endeavor to synthesize bulk sound-absorbing materials using a bottom-up approach. It is particularly noteworthy that our approach does not rely on traditional metal support structures, which are often used in the development of acoustic materials. Instead, we synthesize these materials directly from GO and PEI, which makes our process unique and highly significant in the field of acoustic material development. This bottom-up synthesis of a two-dimensional nanomaterial like GO with a polymer such as PEI, specifically targeting the high-frequency range of 1000-6400 Hz, underscores the novelty and importance of our work. We believe that the optimization strategies employed in our study, along with the resulting materials, provide a new avenue for addressing the challenges associated with high-frequency noise mitigation, especially in scenarios where conventional materials fall short.
Comments No. 3: The details of the material samples(graphene foam and melamine foam) and the experiment setup need to be provided to justify the comparison results shown in Fig.5.
Our response No. 3: Thank you for pointing out this issue. We understand the significance of including comprehensive material and experimental details to substantiate the comparison results presented in Fig. 6 of the revised manuscript. In this study, the sound absorption coefficient was determined according to the KS F 2814-2 standard, which aligns with the transfer function method specified in the international standard ISO 10534-2, thereby ensuring compliance with Korean industrial standards. Further elaboration on the measurement methodology can be found in the section discussing Fig. 6 of the revised manuscript. The dimensions of the sample mold were 29 mm in diameter and 50 mm in thickness, while the GPF (1:3) sample was fabricated with dimensions of 29 mm in diameter and 46 mm in thickness for the sound absorption test. To ensure consistency in comparison, the melamine foam samples were also prepared using the same dimensions. The images shown below, which provide a visual reference of the sample preparation process, have been included in Supplementary Materials of the revised manuscript to adequately address the reviewer's concerns and enhance the clarity of our experimental framework. (Supplementary Materials, Figure S1)
Figure 3. Images of the mold used for GPF fabrication (a) and the GPF (1:3) (b) and the melamine foam (c) samples utilized for sound absorption measurements. (Figure 3 has been included in Supplementary Materials of the revised manuscript as Figure S1)
References for comments of Reviewer #2
- Sujon, M. A. S.; Islam, A.; Nadimpalli, V. K. Damping and Sound Absorption Properties of Polymer Matrix Composites: A Review. Polym. Test. 2021, 104, 107388. https://doi.org/10.1016/j.polymertesting.2021.107388.
- L. Johnson, J. Koplik, R. Dashen, Theory of dynamic permeability andtortuosity in fluid saturated porous media, J. Fluid Mech. 176 (1987) 379–402,https://doi.org/10.1017/S0022112087000727.
- F. Allard, Y. van, New empirical equations for sound propagation in rigid framefibrous materials, J. Acoust. Soc. Am. 91 (1992) 3346–3353, https://doi.org/10.1121/1.402824.
- F. Allard, B. Castagnede, M. Henry, W. Lauriks, Evaluation of tortuosity inacoustic porous materials saturated by air, Rev. Sci. Instrum. 65 (1994) 754–755,https://doi.org/10.1063/1.1145097.
- Chevillotte, C. Perrot, Effect of the three-dimensional microstructure on thesound absorption of foams: a parametric study, J. Acoust. Soc. Am. 142 (2017)1130–1140, https://doi.org/10.1121/1.4999058.
- Sujon, M. A. S.; Islam, A.; Nadimpalli, V. K. Damping and Sound Absorption Properties of Polymer Matrix Composites: A Review. Polym. Test. 2021, 104, 107388. https://doi.org/10.1016/j.polymertesting.2021.107388.
- E. Delany, E.N. Bazley, Acoustical properties of fibrous absorbent materials,Appl. Acoust. 3 (1970) 105–116, https://doi.org/10.1016/0003-682X(70)90031-9.
- Miki, Acoustical properties of porous materials-Modifications of Delany-Bazleymodels-, J. Acoust. Soc. Japan. 11 (1990) 19–24, https://doi.org/10.1250/AST.11.19.
- Xue, L., Zhang, D., Xiaokaiti·Yibulayin, Wang, T., & Shou, X. (2014). Effects of high frequency noise on female rat's multi-organ histology. Noise & health, 16(71), 213–217. https://doi.org/10.4103/1463-1741.137048
- Liu, C. S., Young, L. H., Yu, T. Y., Bao, B. Y., & Chang, T. Y. (2016). Occupational Noise Frequencies and the Incidence of Hypertension in a Retrospective Cohort Study. American journal of epidemiology, 184(2), 120–128. https://doi.org/10.1093/aje/kwv333
- Abbink, V.; Landes, D.; Altinsoy, M.E. Experimental Determination of the Masking Threshold for Tonal Powertrain Noise in Electric Vehicles. Acoustics 2023, 5, 882-897. https://doi.org/10.3390/acoustics5040051
- Li, X.; Yu, X.; Wei Chua, J.; Zhai, W. Harnessing Cavity Dissipation for Enhanced Sound Absorption in Helmholtz Resonance Metamaterials. Materials Horizons 2023, 10 (8), 2892–2903. https://doi.org/10.1039/D3MH00428G.
- Hamdan, N. I.; Zainulabidin, M. H. Sound Absorption Characteristics of Integrated Membrane-Fabric Materials. Journal of Advanced Research in Applied Mechanics 2024, 113 (1), 63–78. https://doi.org/10.37934/aram.113.1.6378.
- Jang, E.-S.; Kang, C.-W.; Kang, H.-Y.; Jang, S.-S. Sound Absorption Property of Traditional Korean Natural Wallpaper (Hanji). Journal of the Korean Wood Science and Technology 2018, 46 (6), 703–712. https://doi.org/10.5658/WOOD.2018.46.6.703.
